# Study on Graphene/CNC-Coated Bamboo Pulp Fabric Preparation of Fabrics with Thermal Conductivity

**DOI:** 10.3390/polym11081265

**Published:** 2019-07-31

**Authors:** Feng Yang, Cuiqin Lan, Haiming Zhang, Jian Guan, Fan Zhang, Benhua Fei, Jilei Zhang

**Affiliations:** 1Fashion Accessory Art and Engineering College, Beijing Institute of Fashion Technology, Beijing 100029, China; 2Key Laboratory of Bamboo and Rattan Science and Technology of the State Forestry Administration, Department of Bio-materials, International Centre for Bamboo and Rattan, Futong Dong Dajie, Chaoyang District, Beijing 100102, China; 3Department of Sustainable Bioproducts, Mississippi State University, Mississippi State, MS 39762, USA

**Keywords:** fabric, graphene, cellulose nanocrystal, thermal conductivity

## Abstract

Functional fabrics have gained attention as an environmentally-friendly synthesis route. In the current study, novelty bamboo pulp fabrics with thermal conductivity properties were prepared by coating the fabric with graphene and cellulose nanocrystal (G/CNC) solutions. The influences of G and CNC concentrations on properties of fabrics were studied. The viscosities of the G/CNC solutions increased with an increase of G contents. G had an obvious thickening effect. Furthermore, compounded fabrics with different G and CNC contents (GCBPFs) were prepared and extensively characterized in terms of thermal and mechanical properties, and morphology. The ultimate thermal conductivity, bursting strength, and tensile strength of the GCBPF were 0.136 W/m·K, 1.514 MPa, and 25.8 MPa, with 4 wt.% CNC and 3 wt.% G contents, respectively. The results demonstrated that the as-fabricated GCBPFs with favorable thermal conductivity could be applied as a novel fast cooling textile for the clothing industry.

## 1. Introduction

Energy consumption was one of the major issues for humans in the 21st century [1,2]. To regulate indoor temperature, energy consumption for space heating and cooling in residential and commercial buildings were dominant, which contributed to 12.3% of total U.S. energy consumption [3]. The concept of “personal thermal management” has emerged as a promising alternative for reducing the demand for indoor temperature regulation. A personal cooling technique could provide thermal comfort by locally cooling the temperature of an individual in a low-cost and energy-saving way [4].

First efforts for personal thermal management techniques to regulate the thermal conductivity of fabrics focused on thickness [5], aerial density [6], porosity [7], and weave structure of the fabric matrix [8]. The combination of personal thermal management with wearable devices was subsequently regarded as one of the most promising strategies, with technologies including cold pack textiles with phase change materials [9], air-adjusted textiles [10], and liquid-adjusted textiles [11]. All the above technologies have their limitations, such as unremarkable performance improvement, inconvenience from the bulky size of the devices, massive consumption of energy, complex craft-work, and high-cost. To address these issues, directly incorporating thermal management materials into textiles for effective personal thermal management has received tremendous attention recently. Hsu et al. [3] reported a good IR-transparent textile made from nanoporous polyethylene, with interconnected pores from 50 to 1000 nm in diameter, for efficient human body cooling. The same group further developed a wearable face mask based on their nanoporous system. Yang et al. [12] demonstrated a scalable randomized glass-polymer hybrid metamaterial for daytime individual thermal regulation.

Bamboo has emerged as the ultimate green, cost-efficient textile material that meets the definition of a renewable and sustainable raw material, with little harmful impact to the environment due to its fast growing speed (3–4 years renewal rate) and abundant quantity in China [13,14,15]. Compared to traditional fibers, the closed-loop system production process from bamboo pulp can be eco-friendly without any harmful substances going into the eco-system [16,17]. Therefore, bamboo fiber as an alternative to traditional fibers has been employed in the production of apparel, sanitary and hygiene products, geo-textiles, composites, and filtration fabrics [18,19].

For fibrous textile materials, conduction, convection, and radiation are the three main routes to processing heat transfer, among which the most significant way is conduction [20]. Therefore, heat conductive textiles for individual thermal regulation can be attractive [21,22,23]. The conductive textiles for personal cooling lead to a satisfactory body feeling due to a decrease in body-generated heat escaping easily into the external environment. Based on the conductive textiles, the concept of direct coating conductive materials onto the conventional textile has been seen as one of the more effective and simple options for cooling textiles, which can remain flexible and wearable [24]. Further, dip-coating methods for cost-effective thermal conductive textiles are attractive because of their similar dying processes to large scale applications in the textile industry [25]. Concerning these issues, various conductive materials, such as iron [26], steel, copper [27] and carbon materials [28] have been utilized to coat textile fibers for making personal cooling conductive textiles [29]. Among these coating materials, graphene (G) has emerged as a revolutionary functional material due to its amazing high thermal conductivity properties (above 3000 W·m·K^−1^) [30]. Coated graphene materials include mainly G, oxidation graphene (GO) and oxidation-reduction graphene (RGO). When graphene is employed in aqueous solutions for coating fabrics, it tends to precipitate due to its no charged surface compared to GO and RGO. For aiding the deposition, dispersants should be used to help G dispersing in stable solutions. Chitosan [31], polyether-imide (PEI) [32], sodium dodecyl sulfate (SDS) [33], sodium cholate surfactant [34], and cellulose fibers [35] have been employed as a dispersant in an attempt to bound G with treated fabrics or textile fibers.

In this study, G material was dispersed in a cellulose nanocrystalline (CNC) aqueous solution to make a G/CNC solution. The solution was coated onto bamboo pulp fabrics to the compound thermal conductive textile by a dip-coating method. The thermal and mechanical properties of the G/CNC-coated bamboo pulp fabrics (GCBPFs) were evaluated by the thermal conductivity, tensile strength, bursting strength, viscosity, and crystallinity.

## 2. Experimental Section

### 2.1. Materials

Plain weave fabric made from 100% bamboo viscose (14.8 gsm, 0.5 mm of thickness) was purchased from Jilin Chemical Fiber Co. Ltd. (Jilin, China). The bamboo fabric was bleached using a standard process to remove inherent impurities and additives. All treated fabrics were subjected to the same scoring and beaching process before coating.

The multilayer graphene-based carbon nano-materials were provided by the Department of Sustainable Bio-products, Mississippi State University (Starkville, Mississippi, US). The graphene nanoplates consisted of several sheets (with a range of 2–30 layers) of graphene with an overall thickness of approximately 1–10 nm [36]. The morphology of graphene material is shown in Figure 1. Microcrystalline cellulose (column chromatography 97%) was obtained from Shanghai Jinsui Biological Inc. (Shanghai, China), and sulfuric acid (analytical reagent 98%) from Nanjing Chemical Reagents Co. (Nanjing, China). The deionized water was used exclusively.

### 2.2. Preparation of Coating Solution

The cellulose nano-crystalline (CNC) suspension was prepared through the sulfuric acid hydrolysis of microcrystalline celluloses [37]. The obtained CNC suspension had the typical acicular structure, with dimensions ranging from 100 to 200 nm in length and 5 to 10 nm in diameter, as shown in Figure 2. There were three concentration levels of CNC suspensions (1, 2, and 4 wt.%) used to disperse G for evaluating the effect of CNC concentrations on the degree of G dispersion. The G and CNC mixing procedure started by adding G materials with 1, 2, and 3 wt.% into CNC suspensions, followed by ultrasonic treatment for 2 min, respectively. Then the G/CNC solutions were obtained with different G and CNC ratios. The additions of G were the ratios to the weight of CNC suspension.

### 2.3. Coating Procedure

The prepared G/CNC solutions were applied to impregnate the bamboo pulp fabrics simply by a dip-coating method. The fabrics were first dipped into the G/CNC solutions and then padded through a rolling device to remove the excessive solutions. The above procedure was repeated for several times until the resistance of the fabric became constant. The obtained G/CNC coated bamboo pulp fabrics (GCBPFs) were dried at 45 °C for 2 h before being tested. For comparison, CNC suspensions coated fabric samples (CBPFs) were also prepared.

### 2.4. Characterizations

For measurement of the viscosity of dipping solution before and after the addition of G, an NDJ-79 rotational viscometer was used with a rotational speed of 750 r·min^−1^. The surface morphologies of G, CNC, and fabrics were tested by Quanta 200 scanning electron microscope (SEM, FEI company, Hillsboro, OR, US) and transmission electron microscope (TEM, FEI company, Hillsboro, OR, US), respectively.

The X-ray diffraction (XRD) was performed using Cu kα radiation source (step size 0.02°, scanning speed 6°/min, voltage 40 kV, current 100 mA) in an Ultima IV XRD instrument (Rigaku company, Tokyo, Japan). The tensile strength of fabrics was determined according to ASTM D2256 by an HY-932CS micro-computer control electron universal testing machine, with a tensile rate of 1 mm/min. The bursting strength of samples was evaluated according to ASTM D3787-07 using a bursting strength tester (HY-953, Hengyu company, Dongguan, China). Five duplicates were repeated in each mechanical test group.

A heat conduction coefficient tester (ISOMET 2104, Judeng company, Beijing, China) was employed to record the heat conductivity of samples. The test standard was according to GB/T 11048-2018. The thermal conductivity (λ) was calculated by using the following law:(1)$$\lambda =mc\frac{\Delta T}{\Delta t}|{}_{T=T_{2}}•\frac{h}{T_{1}-T_{2}}•\frac{1}{\pi R^{2}}$$
where λ is the thermal coefficient (W/m·k), m is the quality of copper cooling plate (kg), c is the specific heat capacity of copper cooling plate, 385 J/(kg·K), ${\frac{\Delta T}{\Delta t}|}_{T=T_{2}}$ is the heat dissipation rate of copper cooling plate at *T*_2_ (mV/s), h is the thickness of tested sample (m), $T_{1}-T_{2}$ is the temperature difference between upper and lower sample surfaces (K), and $\pi R^{2}$ is the area of copper cooling plate (m^2^). 

## 3. Results and Discussion

### 3.1. Viscosity of G/CNC Solution

The viscosity, a key feature for the fabric impregnation solution, depends on the solid content, pH value, and temperature of finishing liquid [38]. For the textile industry, it is difficult to impregnate finishing liquid into fiber bundles because of the extremely high viscosity of impregnation solutions or resins. Figure 3a shows the viscosities of G/CNC solutions with different ratios of G to CNC suspension. For the pure CNC suspensions, viscosity increased with increasing CNC contents in the suspensions. Specifically, the viscosity of 4 wt.% CNC suspension was 226.7 mPa s, which was much higher than that of the 1 wt.% CNC suspension (20.1 mPa s). A three-dimensional network structure was formed through hydrogen bonding between each of the other groups (hydroxyl, carbonyl or carboxyl) on CNC or the groups with water. The network structure was constantly improved and strengthened with an increasing amount of CNC [39]. Moreover, the viscosities of the G/CNC solutions increased with an increase of G contents. It was found that G had an obvious thickening effect. The viscosity of the G/CNC solution appeared to have a maximum value of 312.6 mPa s with the concentrations of 3 wt.% G and 4 wt.% CNC. There were many factors that affected the rheological properties of polymers, such as polymer chain structure, relative molecular weight, and solution concentration. The viscosity of the majority of polymers increased with an increase of relative molecular weight or solution concentration. Figure 3b shows the viscosity increase rate of the G/CNC solution, which was calculated as the viscosity difference between G/CNC solution and pure CNC suspension divided by the viscosity of pure CNC suspension. The G/CNC solution with 3 wt.% G and 1 wt.% CNC suspension had the highest viscosity increase rate of 108.5%, and the lowest value appeared at the solution with 1 wt.% G and 4 wt.% CNC suspension.

### 3.2. XRD Analysis

The XRD test was carried out to investigate the degree of crystallinity (*Cr*) for the pure CNC suspension and G/CNC solutions. The *Cr* of pure CNC suspension was calculated at 83%, and three concentration levels of CNC suspensions (1, 2, and 4 wt.%) had the same *Cr* values as shown in Figure 4. Simultaneously, it was found that the *Cr* values of the G/CNC solutions were lower than that of pure CNC, and decreased with an increase of G contents. The lower the CNC suspension concentrations, the faster the *Cr* values descend in range of G/CNC solutions with the increase of G contents. The lowest *Cr* value of the G/CNC solution was 19% with 3 wt.% G and 1 wt.% CNC. This phenomenon may be the continuous crystallization of CNC in the G/CNC solution which was organized by graphene sheets [40], resulting in a significant decrease in the peak value of *Cr* in the G/CNC dispersion. In addition, the interaction between graphene and CNC may also lead to the disappearance of the periodic interval between graphene layers, thus greatly improving the dispersion of graphene in CNC suspension [41], and decreasing the *Cr* of G/CNC solution. From the TEM images (Figure 5a), it can be seen that graphene sheets were homogeneously coated well on the surface of CNC via a hydrogen bonding interaction and the rod-like morphology, and the size of CNC was retained well after the reaction [42]. Uniform and stable G/CNC solutions were formed, which was left at room temperature for 3 months without stratification or precipitation as shown in Figure 5b.

Figure 6 shows the XRD pattern of CNC and G/CNC-coated fabrics with the lowest and highest thermal conductivities. It can be observed in the curve of CBPF that the diffraction peaks appeared around 2θ = 12.2° and 21.7°, respectively, which is the typical cellulose II crystal structure. The diffraction peaks decreased to 2θ = 21.2° and 21.5° in GCBPF-L and GCBPF-H, respectively. The GCBPFs had very similar characteristic peaks with CBPF. As shown in Figure 6, the addition of graphene resulted in the decrease of *Cr* for the G/CNC solutions, therefore the obtained composite fabrics (GCBPFs) also had the lower diffraction peak values than the CBPF one.

### 3.3. Thermal Conductivity

The thermal conductivities of CBPFs ranged from 0.046 to 0.049 W/m·K tested by the thermal conductivity meter using a steady plate method as shown in Figure 7. The G/CNC-coated fabrics exhibited better conductive behavior than the pure CNC-coated samples. Due to the existence of the graphene in the G/CNC solutions, the solution coated bamboo pulp fabric turned into thermal conductive functional fabrics (GCBPFs). During the test, the thermal conductivity of all GCBPF samples had been improved, indicating that the addition of graphene can further improve the thermal conductivity of fabric to a certain extent. The maximum thermal conductivity of the GCBPF was 0.136 W/m·K, which was 178–196% higher than the CBPF sample, and as high as some previously prepared graphene coated conductive viscose fabrics [18,19]. After dispersing by CNC suspension uniformly, the lamellar structure of G can be maintained by van der Waals or restoration π-π stacking between adjacent G layers.

The GCBPF sample with a higher CNC concentration of 4 wt.% had higher thermal conductivity with different graphene contents from 1 wt.% to 3 wt.%. Lower thermal conductivity for low contents of CNC might be due to situations such as the coating solution (to some extent) blocking the interstices and pores, and water droplets spreading in/on the fabric [43]. Furthermore, energy came from the effective dissipation of heat through the different coated layer referring irradiative, and it was difficult to be measured. To clear the thermal conductive property systematically, the investigation of air-permeability, wettability, and color appearance of the graphene-coated fabric had been arranged for further study.

### 3.4. Tensile and Bursting Properties

The results of tensile and elongation at break were plotted in Figure 8 and Figure 9, respectively. When impregnated with pure CNC suspension, the elongation at break decreased a little with an increase of CNC contents from 1 to 4 wt.%. The addition of graphene improved the elongation at break of G/CNC solution coated fabrics, and with the increase of graphene content the elongation at break increased. The maximum elongation at break for GCBPF appears in 3 wt.% G and 2 wt.% CNC, with the value of 12.5%. As shown in Figure 8, the elongation at break increased with an increased content of graphene. The addition of graphene destroyed the crystallization of the CNC solution, and the flake graphene increased the flexibility of the G/CNC impregnating solution, which resulted in the decrease of *Cr* (Figure 4) and increase of elongation at break. 

A gradual increase in the tensile strength (TS) of CPBFs was observed (from 24.9 MPa to 29.0 MPa) in Figure 9 when the CNC concentration was increased from 1 wt.% to 4 wt.%. Moreover, the TS of GCBPFs decreased significantly (*p* < 0.05) with the addition of graphene from 1 wt.% to 3 wt.% compared with CPBF, with the values of TS in GCBPFs ranging from 18.2 MPa to 25.8 MPa. The larger decrease of TS in GCBPFs than CBPFs might be contribute to the addition of graphene destroying the complete self-assembly of CNC and the stronger electron scattering by graphene resulting in GCBPFs yielding brighter structures than CBPFs [44].

The bursting strength (BS) of CPBFs and GCPBFs is shown in Figure 10. The values of bursting strength in CPBFs increased with an increase of CNC contents, while the addition of graphene made the values decrease significantly (*p* < 0.05). The ultimate strength of the GCPBF was up to 1.514 MPa, which was slightly lower than the CPBF (1.696 MPa). This indicates that GCPBFs inherited the excellent mechanical properties of CPBFs, and the dip coating treatment did not change the inner structure of fibers; the complex of graphene and hydroxyl groups in cellulose makes cellulose molecules cross-linked [45]. Meanwhile, the GCPBFs also had excellent thermal conductivity mentioned by the former discussion in this paper.

### 3.5. Morphology of Coated Bamboo Pulp Fabrics

The structures of the fabrics and G/CNC-coated bamboo viscose fabrics (GCBPFs) were investigated by SEM as shown in Figure 11. It can be seen from Figure 11a,b that pristine bamboo viscose fabric (BPF) had smooth surfaces, while the GCBPFs had much coarser surfaces as shown in Figure 11c–f, indicating the G/CNC solutions had successfully coated the bamboo viscose fiber surface.

Figure 11c,d are images of GCBPFs with 1 wt.% CNC and 1 wt.% G, which had the lowest thermal conductivity. With increasing contents of CNC and G by 4 wt.% and 3 wt.%, respectively, the graphene can be seen to be uniformly distributed on the surface of the fabrics (Figure 11e,f) and interacting with the bamboo fibers through hydrogen interactions. The GCBPFs are filled more tightly by the G/CNC fillers, which also correspond to highest thermal conductivity [46]. 

## 4. Conclusions

In summary, the GCBPFs were synthesized by impregnating BPFs into the G/CNC solutions through a wet coating process in this study. The structure, and mechanical and thermal properties were studied subsequently. The graphene was found to uniformly coat bamboo pulp fiber surfaces due to the dispersion of CNC suspension, which enhanced the thermal and mechanical properties of the fabric at the same time. The GCBPF sample with a higher CNC concentration of 4 wt.% and G of 3 wt.% had excellent thermal conductivity and mechanical properties. The G/CNC coatings formed a firmly linked thermal conductive network, which made the originally insulating BPF thermally conductive. The results of this study indicate that the prepared GCBPFs have great application potentials for cooling textiles, conductive fabrics, wearable electronics, etc. Moreover, the durability of the G/CNC coating, i.e., the washing experiment, color fastness, and wear test, will be evaluated in order to validate the practical interest of this solution in our future work.

## Figures and Tables

**Figure 1 polymers-11-01265-f001:**
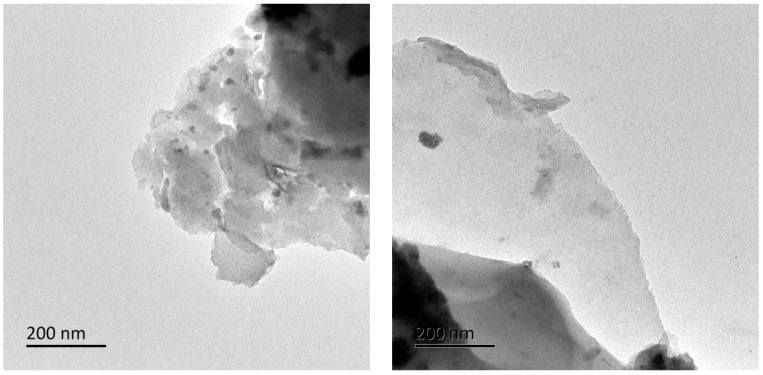
TEM images of graphene materials.

**Figure 2 polymers-11-01265-f002:**
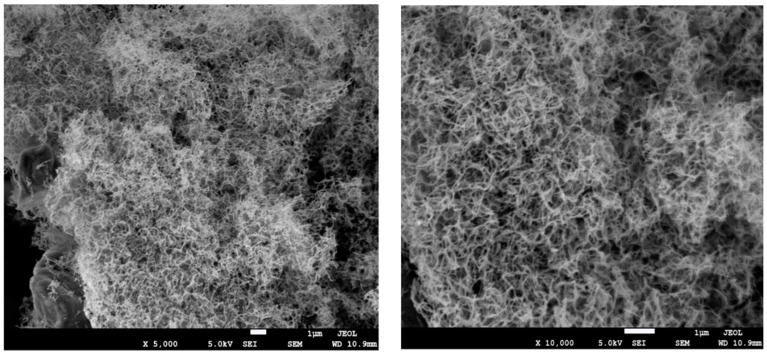
SEM images of cellulose nanocrystalline (CNC).

**Figure 3 polymers-11-01265-f003:**
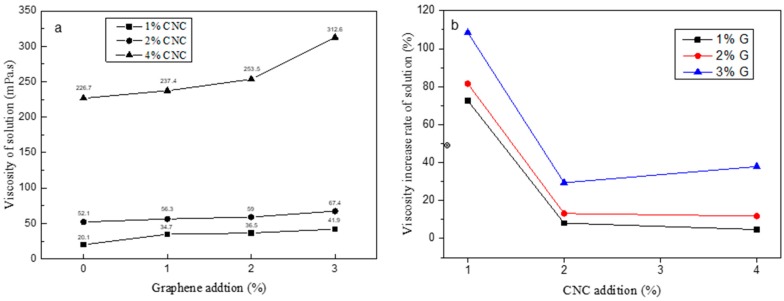
Viscosity (**a**) and viscosity increase rate (**b**) of the G/CNC solution.

**Figure 4 polymers-11-01265-f004:**
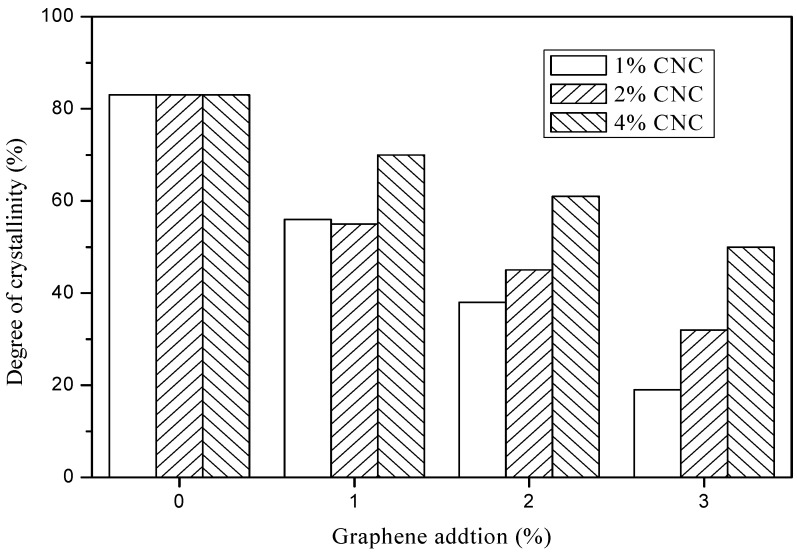
Degree of crystallinity of G/CNC solutions.

**Figure 5 polymers-11-01265-f005:**
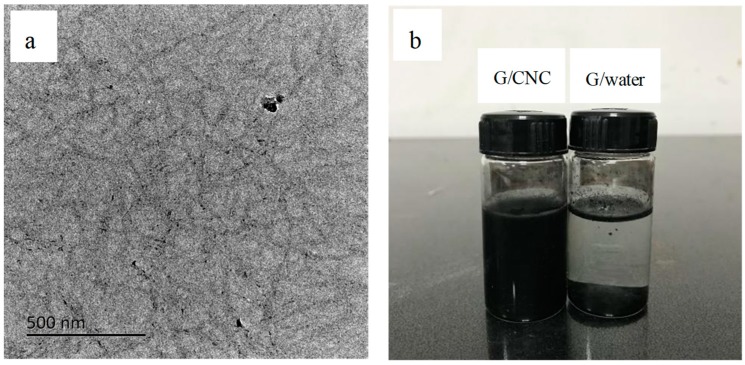
TEM image of G/CNC solution with 3 wt.% G and 4 wt.% CNC (**a**) and photo of graphene/ CNC suspension and water after 3 months at room temperature (**b**).

**Figure 6 polymers-11-01265-f006:**
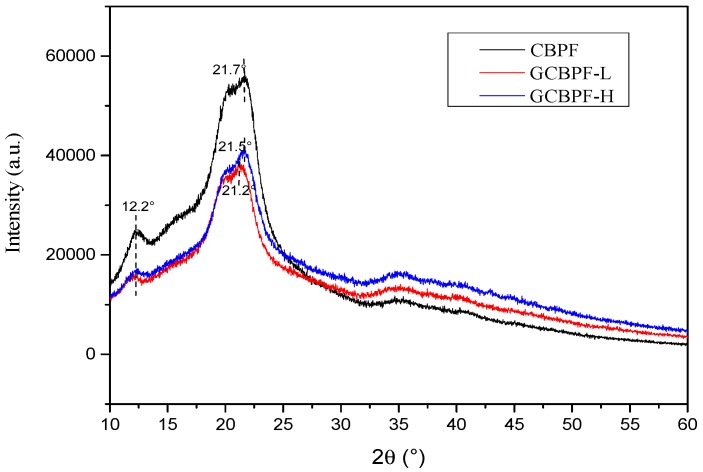
Degrees of crystallinity of CNC-coated bamboo pulp fabric (CBPF), and G/CNC-coated fabrics with the lowest (GCBPF-L) and highest conductivities (GCBPF-H).

**Figure 7 polymers-11-01265-f007:**
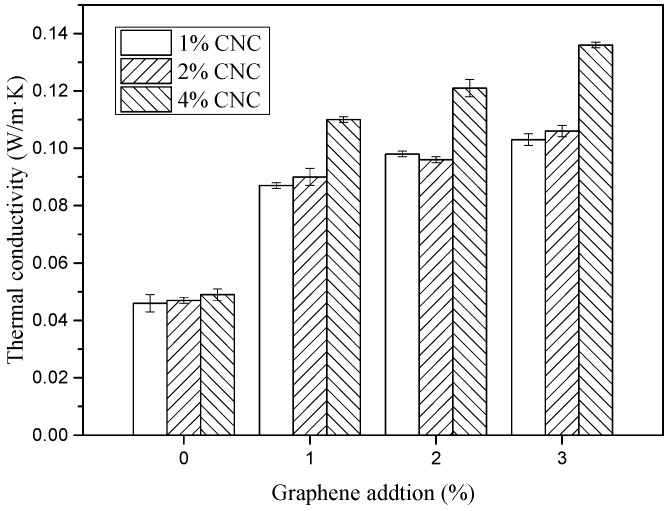
Thermal conductivities of CBPFs and GCBPFs.

**Figure 8 polymers-11-01265-f008:**
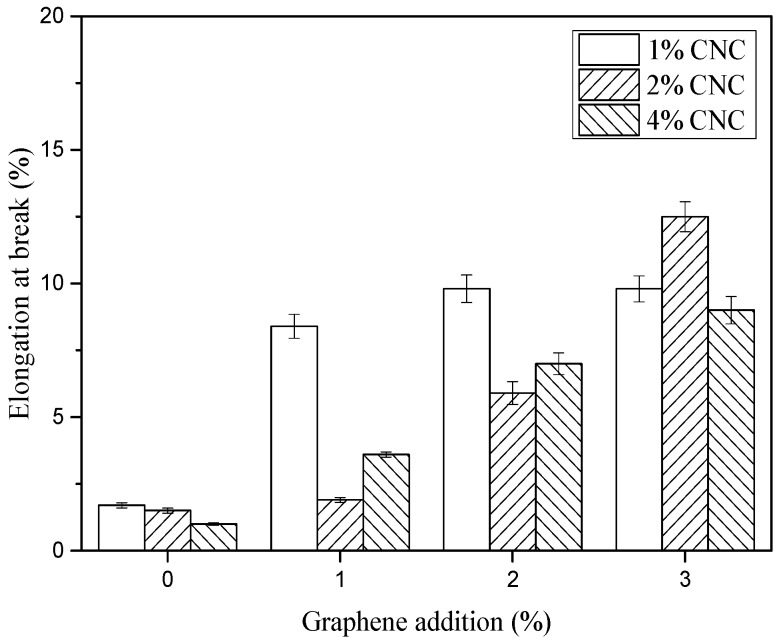
Elongation at break of CBPFs and GCBPFs.

**Figure 9 polymers-11-01265-f009:**
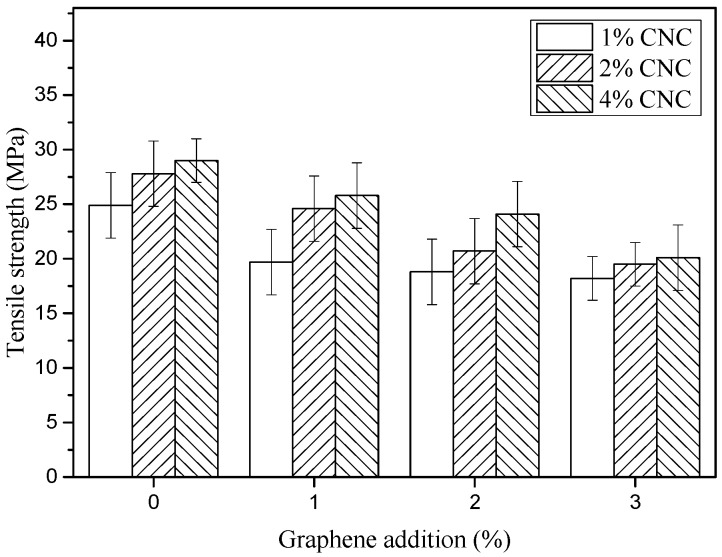
Tensile strength of CBPFs and GCBPFs.

**Figure 10 polymers-11-01265-f010:**
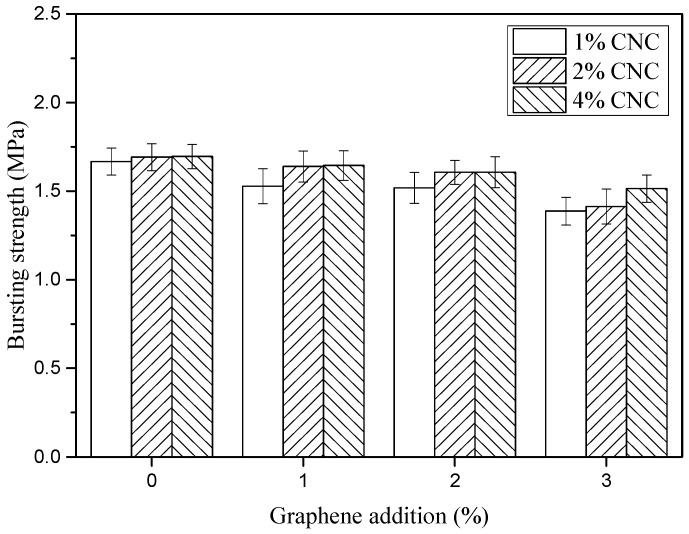
Bursting strength of CBPFs and GCBPFs.

**Figure 11 polymers-11-01265-f011:**
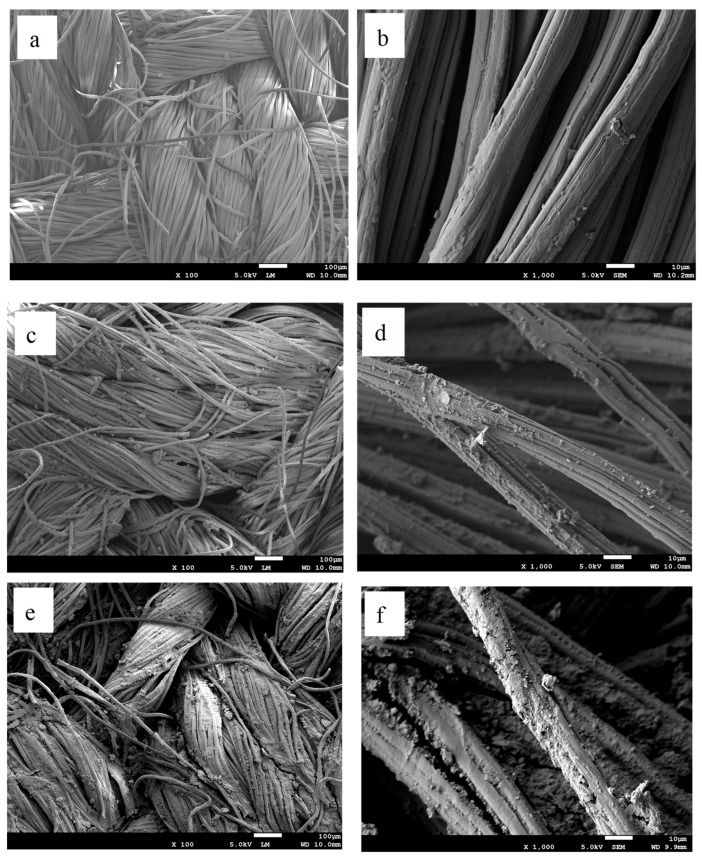
SEM images of CBPFs and GCBPFs ((**a**,**b**) CNC-coated bamboo pulp fabric; (**c**,**d**) G/CNC-coated fabric with lowest thermal conductivity; (**e**,**f**) G/CNC-coated fabric with highest thermal conductivity).

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
