# Peer review of "Study on Graphene/CNC-Coated Bamboo Pulp Fabric Preparation of Fabrics with Thermal Conductivity"

_polymers, 2019, doi:10.3390/polym11081265_

Reviewer 1 Report

This manuscript by Yang et al. demonstrates a bamboo pulp fabric coating made from graphene/cellulose nanocrystal (CNC) solutions via a dip-coating method. The simplicity of the coating fabrication onto the fabrics makes this an interesting work worthy of publication. The combination of graphene and CNC exhibits a synergistic effect on thermal conductivity, tensile strength, and mechanical properties and has a clear application. This coating seems have good potential for real world application including cooling textiles, wearable electronics, etc.

A few comments;

1) Although overall the English of the manuscript is good, there are a few small places where incorrect prepositions are used; the authors should check over their writing carefully.

2) In page 5 (lines 160-162), it needs to be rephrased.

3) In page 6 line 166, how would a hydrogen bonding interaction be possible for graphene and CNC? I think that the authors’ explanation for this is not reasonable. That’s a little unclear to me. Where is this intermolecular interaction originated from? Assuming that graphene is completely made of “Carbon”, there wouldn’t be no hydrogen bonding interactions between two components. I wonder if the authors are missing some important information about graphene surfaces.

4) In the experimental (coating procedure in page 3) the authors state that the dipping procedure was repeated several times. Have the authors tried to measure weight gains or thickness increment with the number of cycles? It would be nice to see any trend of mechanical and thermal properties with deposition cycles as the authors studied them with graphene contents.

Author Response

Dear Editor and Reviewers: I am pleased to resubmit for the revised version of manuscript entitled “Study on graphene/CNC-coated bamboo pulp fabric part I: preparation of fabrics with thermal conductivity” (ID: polymers-532415). Thank you for reading our manuscript and reviewing it. Those comments are all valuable and very helpful for revising and improving our paper. We have revised our manuscript carefully and have made correction which we hope meet with approval. So, we have sent the revised manuscript and have highlighted changes by using the yellow colour.

The main corrections in the paper and the responds to the reviewers’ comments are as following: Responds to the reviewers’ comments:

Reviewer 1 Comments:

1) Although overall the English of the manuscript is good, there are a few small places where incorrect prepositions are used; the authors should check over their writing carefully.

Answer: The language of this manuscript was proofread.

2) In page 5 (lines 160-162), it needs to be rephrased.

Answer: The sentence “The XRD was carried out for the pure CNC suspension and G/CNC solutions to investigate the degree of crystallinity (Cr). Cr of pure CNC suspension is calculated by 83 % and different concentrations had the same Cr values (Figure 4)” was rewritten by “The XRD test was carried out to investigate the degree of crystallinity (Cr) for the pure CNC suspension and G/CNC solutions. The Cr of pure CNC suspension was calculated by 83 % and three concentration levels of CNC suspensions (1, 2, and 4 wt%) had the same Cr values as shown in Figure 4”.

3) In page 6 line 166, how would a hydrogen bonding interaction be possible for graphene and CNC? I think that the authors’ explanation for this is not reasonable. That’s a little unclear to me. Where is this intermolecular interaction originated from? Assuming that graphene is completely made of “Carbon”, there wouldn’t be no hydrogen bonding interactions between two components. I wonder if the authors are missing some important information about graphene surfaces.

Answer: Wang et al. (2018) reported naked Au nanoparticles monodispersed onto multifunctional cellulose nanocrystals-graphene hybrid sheets and found CNC were homogeneously coated well on the surface of graphene sheets via a hydrogen bonding interaction and the rod-like morphology and size of the CNC in hybrid sheets could been retained well after reaction ([41] Wang Y, Zhang H, Lin X, Chen S, Jiang Z, Wang J, Huang J, Zhang F, Li H. Naked Au nanoparticles monodispersed onto multifunctional cellulose nanocrystals-graphene hybrid sheets: Towards efficient and sustainable heterogeneous catalysts [J]. New Journal of Chemistry, 2018, 42: 2197-2203.). Our graphene supplied by Mississippi State University (Mississippi, US), with the properties of the graphene nanoplates consisting of several sheets (with a range of 2-30 layers) of graphene with an overall thickness of approximately 1-10 nm ([36] Yan Q, Zhang X, Li J, Hassan E B, Wang C, Zhang J, Cai Z. Catalytic conversion of Kraft lignin to bio-multilayer graphene materials under different atmospheres [J]. Journal of Materials Science, 2018, 53: 8020.). In the FTIR test, there is a strong and wide stretching vibration peak of -OH in the range of 3000- 3700 cm-1, and the C-O peak appeared at 1055 cm-1, indicating that our graphene partly retained the unconverted kraft lignin (our raw materials for making graphene).

4)In the experimental (coating procedure in page 3) the authors state that the dipping procedure was repeated several times. Have the authors tried to measure weight gains or thickness increment with the number of cycles? It would be nice to see any trend of mechanical and thermal properties with deposition cycles as the authors studied them with graphene contents.

Answer: The above procedure was repeated for several times until the resistance of the fabric becoming constant. Thanks for the excellent suggestion of “the effects deposition cycles and graphene contents on the mechanical and thermal properties ”, the relative content will be investigated in our further study. We appreciate for Editor and Reviewers’ warm work earnestly, and hope that the correction will meet with approval. Once again, thank you very much for your comments and suggestions.

Yours sincerely,

Corresponding author: Name: Feng Yang; Benhua Fei E-mail: yangfeng@bift.edu.cn; feibenhua@icbr.ac.cn

Reviewer 2 Report

Manuscript by Yang et al. Describes the effect of graphene and cellulose nanocrystals on mechanical and thermal properties of  bamboo pulp. The manuscript is clearly written and the conclusions are supported by the results. I have the following minor points, to be added before publication.

1-In Figure 3b and the text related to the figure, the definition of viscosity increase rate is not clear. Please clarify this point.

2-Figure 4, normally addition of particles such as graphene increases the degree of crystallization of polymers (see for example J. Phys. Chem. B 113, 5568, 2009). However the results in Figure 4 indicate a decrease in crystallinity with increasing the graphene content. Is there any special reason about that?

p { margin-bottom: 0.1in; line-height: 115%; }a:link { }

Author Response

I am pleased to resubmit for the revised version of manuscript entitled “Study on graphene/CNC-coated bamboo pulp fabric part I: preparation of fabrics with thermal conductivity”. Thank you for reading our manuscript and reviewing it. Those comments are all valuable and very helpful for revising and improving our paper. We have revised our manuscript carefully and have made correction which we hope meet with approval. So we have sent the revised manuscript and have highlighted changes by using the yellow colour. The main corrections in the paper and the responds to the reviewers’ comments are as following:

1-In Figure 3b and the text related to the figure, the definition of viscosity increase rate is not clear. Please clarify this point.

Answer: Figure 3 (b) showed the viscosity increase rate of the G/CNC solution, which was calculated as the viscosity difference between G/CNC solution and pure CNC suspension divided by the viscosity of pure CNC suspension.

2-Figure 4, normally addition of particles such as graphene increases the degree of crystallization of polymers (see for example J. Phys. Chem. B 113, 5568, 2009). However the results in Figure 4 indicate a decrease in crystallinity with increasing the graphene content. Is there any special reason about that?

Answer: This phenomenon may be the continuous crystallization of CNC in the G/CNC solution was organized by graphene sheets ([40] Carrasco P M, Montes S, García I, Borghei M, Jiang H, Odriozola, I. High-concentration aqueous dispersions of graphene produced by exfoliation of graphite using cellulose nanocrystals [J]. Carbon, 2014, 70(4): 157-163.), which resulted in a significant decrease in the peak value of Cr in the G/CNC dispersion. In addition, the interaction between graphene and CNC may also lead to the disappearance of the periodic interval between graphene layers, thus greatly improving the dispersion of graphene in CNC suspension ([41] Wang Y, Zhang H, Lin X, Chen S, Jiang Z, Wang J, Huang J, Zhang F, Li H. Naked Au nanoparticles monodispersed onto multifunctional cellulose nanocrystals-graphene hybrid sheets: Towards efficient and sustainable heterogeneous catalysts [J]. New Journal of Chemistry, 2018, 42: 2197-2203.) and decreasing the Cr of G/CNC solution.

We appreciate for Editor and Reviewers’ warm work earnestly, and hope that the correction will meet with approval. Once again, thank you very much for your comments and suggestions.

Yours sincerely,

Name: Feng Yang; Benhua Fei

E-mail: yangfeng@bift.edu.cn; feibenhua@icbr.ac.cn
